# Endogenous bioluminescent reporters reveal a sustained increase in utrophin gene expression upon EZH2 and ERK1/2 inhibition

Hannah J. Gleneadie[1,8], Beatriz Fernandez-Ruiz[1,8], Alessandro Sardini[2], Mathew Van de Pette[1,6], Andrew Dimond [1], Rab K. Prinjha [3], James McGinty[4], Paul M. W. French [4], Hakan Bagci[5], Matthias Merkenschlager [5] & Amanda G. Fisher [1,7✉]

Duchenne muscular dystrophy (DMD) is an X-linked disorder caused by loss of function mutations in the dystrophin gene (*Dmd*), resulting in progressive muscle weakening. Here we modelled the longitudinal expression of endogenous *Dmd*, and its paralogue *Utrn*, in mice and in myoblasts by generating bespoke bioluminescent gene reporters. As utrophin can partially compensate for *Dmd*-deficiency, these reporters were used as tools to ask whether chromatin-modifying drugs can enhance *Utrn* expression in developing muscle. Myoblasts treated with different PRC2 inhibitors showed significant increases in *Utrn* transcripts and bioluminescent signals, and these responses were independently verified by conditional *Ezh2* deletion. Inhibition of ERK1/2 signalling provoked an additional increase in *Utrn* expression that was also seen in *Dmd*-mutant cells, and maintained as myoblasts differentiate. These data reveal PRC2 and ERK1/2 to be negative regulators of *Utrn* expression and provide specialised molecular imaging tools to monitor utrophin expression as a therapeutic strategy for DMD.

[1] Epigenetic Memory Group, MRC London Institute of Medical Sciences (LMS), Imperial College London, Du Cane Road, London W12 0NN, UK. [2] Whole Animal Physiology and Imaging Facility, MRC LMS, Imperial College London, Du Cane Road, London W12 0NN, UK. [3] Immunology and Epigenetics Research Unit, Research, GlaxoSmithKline, Gunnels Wood Road, Stevenage, Herts SG1 2NY, UK. [4] Photonics Group, Department of Physics, Blackett Laboratory, Imperial College London, London SW7 2AZ, UK. [5] Lymphocyte Development Group, MRC LMS, Imperial College London, Du Cane Road, London W12 0NN, UK. [6] Present address: MRC Toxicology Unit, Gleeson Building, Tennis Court Road, Cambridge CB2 1QR, UK. [7] Present address: Department of Biochemistry, University of Oxford, South Parks Road, OX1 3QU Oxford, UK. [8] These authors contributed equally: Hannah J. Gleneadie, Beatriz Fernandez-Ruiz. ✉email: amanda.fisher@lms.mrc.ac.uk

Dystrophin (Dmd/DMD) and utrophin (Utrn/UTRN) are highly related gene paralogues that are located on mammalian X chromosomes and autosomes, respectively[1–4]. Mutations in the human DMD gene that cause dystrophin protein loss result in severe and progressive muscle wasting[5]. Duchenne muscular dystrophy (DMD) is one of the most common X-linked human diseases affecting between one every 3500 to 5000 male births worldwide[6,7]. In patients, loss of dystrophin compromises the structure and function of the sarcolemma and drives unproductive cycles of muscle degeneration and regeneration, as well as inflammation, that collectively culminate in the replacement of muscle with fibrous tissue[5,8]. Although there is currently no effective cure for DMD, treatment with corticosteroids is used to retard disease progression[9] and a range of promising new gene therapy-based and epigenetic approaches are in development or have recently gained medicines agency approval. These focus on exogenous introduction of minimal dystrophin expression cassettes[10–12] or aim to circumvent deleterious mutations either by exon-skipping[13–15] or the prevention of nonsense mediated decay[16,17]. An alternative therapeutic approach relies on harnessing and re-purposing the dystrophin paralogue utrophin[18]. Dystrophin and utrophin are remarkably similar in terms of protein sequence and domain structure, and share the capacity to bind to the dystrophin-associated glycoprotein complex. Functional studies in the mouse mdx model of DMD[19] have shown that utrophin expression can prevent development of the mdx phenotype and improve muscle function[20–22]. Likewise, in a dystrophin-deficient golden retriever model, expression of a miniaturised utrophin transgene in neonatal or 7.5 week old pups prevented development of myonecrosis[23]. These findings have fuelled a search for agents and mechanisms that can enhance utrophin expression[3,24–26].

In skeletal muscle, utrophin is expressed throughout the sarcolemma during foetal and perinatal development, but becomes restricted to neuromuscular and myotendinous junctions postnatally, while dystrophin occupies the sarcolemma[27–29]. Detailed analyses of utrophin gene structure revealed at least two promoters (A and B) that generate different isoforms with distinct expression patterns[30–32]. Since utrophin-A promoter activity correlates most closely with dystrophin expression, much attention has been focused on defining strategies and screening for compounds that enhance Utrn expression via increased Utrn-A promoter activity. While this strategy has enabled the discovery of several utrophin-modulating compounds and transactivating factors that enhance UTRN/Utrn expression[3,33–37], employing artificial constructs that do not fully encompass the endogenous UTRN/Utrn locus could limit the sampling of complex long-range epigenetic mechanisms that operate in vivo to exquisitely shape the developmental expression of utrophin mRNA and proteins. Indeed, as yet, no utrophin upregulating compound has successfully made it to the clinic.

To address this gap, we developed a preclinical mouse model in which the allelic expression of endogenous Dmd and Utrn genes can be simultaneously visualised in vivo. This enables longitudinal monitoring of expression in foetal development, adolescence and adulthood. Myoblast cell lines derived from embryonic and adult reporter mice were used to screen for candidate chromatin-modifying drugs or signalling pathway inhibitors that enhanced endogenous Utrn expression. We show that treatment of myoblasts with PRC2 inhibitors, or conditional genetic deletion of the PRC2 core component Ezh2, enhanced Utrn expression. Treatment with ERK1/2 inhibitors, known to promote myotube fusion via the activation of CaMKII[38], also resulted in a substantial upregulation of Utrn. In contrast, exposure to drugs that disrupt other features of repressive or inactive chromatin, such as DNA methylation and HDAC activity, did not alter Utrn expression. These data provide fresh evidence that epigenetic mechanisms can be selectively harnessed to promote utrophin expression in adult muscle and offer a proof of principle that the bioluminescent reporter mice and myoblast lines described herein can be used to expedite the search for therapeutic routes to tackle DMD.

## Results

**Generating mouse reporters to visualise endogenous Dmd and Utrn gene expression.** To generate mouse lines that accurately report dystrophin and utrophin expression in vivo, we separately targeted a click beetle-derived green luciferase (CBG99Luc) or a red-emitting firefly luciferase (RFluc) and β-galactosidase (lacZ) into the 3′-untranslated regions of endogenous Dmd ($Dmd^G$) or Utrn ($Utrn^R$), respectively, in mouse ESCs (Fig. 1a, b shows sites of insertion relative to exons for each locus). Inserted constructs were engineered with T2A self-cleaving peptide sequences so that expression of the targeted genes (Dmd or Utrn) is not disrupted and anticipated to result in co-ordinated expression of luciferase (or β-galactosidase) as reporters. A similar approach was previously used by our group to study the expression of endogenous imprinted and non-imprinted genes in ESCs and in mice[39–41], and prior studies also suggest that this strategy could enable the study of multiple genes simultaneously[42]. $Dmd^G$ and $Utrn^R$ insertions were separately engineered in ESCs, verified by DNA sequencing, and individual mouse lines were created from targeted ESCs.

$Dmd$-CBG99Luc ($Dmd^G$) mice were imaged using an IVIS spectrum instrument that provides whole-body scans of luciferase activity. Following injection of D-Luciferin substrate, strong bioluminescent signals (green/yellow) were observed in all three $Dmd^{G+/−}$ heterozygote animals as compared with a wildtype (WT) littermate ($Dmd^{G−/−}$) (Fig. 1c). In these reporter mice, bioluminescence signal was prominent in the head, fore and hind limbs, and quantitative analysis confirmed significant radiance (flux) in $Dmd^G$ reporter male and female animals, as compared to matched wildtype controls (Fig. 1d). Among female mice, homozygous animals ($Dmd^{G+/+}$) showed higher levels of bioluminescence than hemizygous ($Dmd^{G+/−}$) animals, as anticipated for an X-linked reporter. Examination of ex vivo tissue isolated from $Dmd^{G+/−}$ mice confirmed bioluminescence signal in the skeletal muscle and to a lesser extent the heart, but not in white adipose tissue (WAT) or kidney, consistent with the predicted expression of Dmd in adult mice (Fig. 1e). Furthermore, molecular analysis of Dmd transcript abundance in each of these tissues indicated that the $Dmd^{G+}$ allele was expressed at a similar level to the wildtype Dmd ($Dmd^{G−}$) allele (Fig. 1f). This suggested that targeting CBG99Luc into the endogenous dystrophin locus had not radically altered the tissue-specific expression of Dmd.

In $Utrn$-RFluc-lacZ ($Utrn^R$) reporter mice, strong bioluminescent signals were detected, particularly in the lower abdomen, that were not seen in the matched wildtype control. Importantly, the distribution of $Utrn^R$ bioluminescent signal was distinct from that seen for $Dmd^G$ animals (Fig. 1g). Quantification of whole-body radiance in male and female $Utrn^R$ and control animals confirmed significantly higher levels in the $Utrn^R$ reporter mice (Fig. 1h) and, consistent with expectations, radiance in homozygotes ($Utrn^{R+/+}$) was two-fold higher than in heterozygotes ($Utrn^{R+/−}$). Tissues isolated from these mice showed prominent bioluminescent signal in WAT and kidney, with a lower signal abundance in heart and skeletal muscle (Fig. 1i). These data also confirmed that tissue-specific expression of utrophin transcripts was indistinguishable between wildtype and $Utrn^R$-targeted alleles (Fig. 1j). The expected tissue-specific distribution of $Utrn^R$ was also verified by assessing the expression of a secondary Utrn reporter, lacZ (Fig. 1k) and by X-Gal staining of adult $Utrn^R$

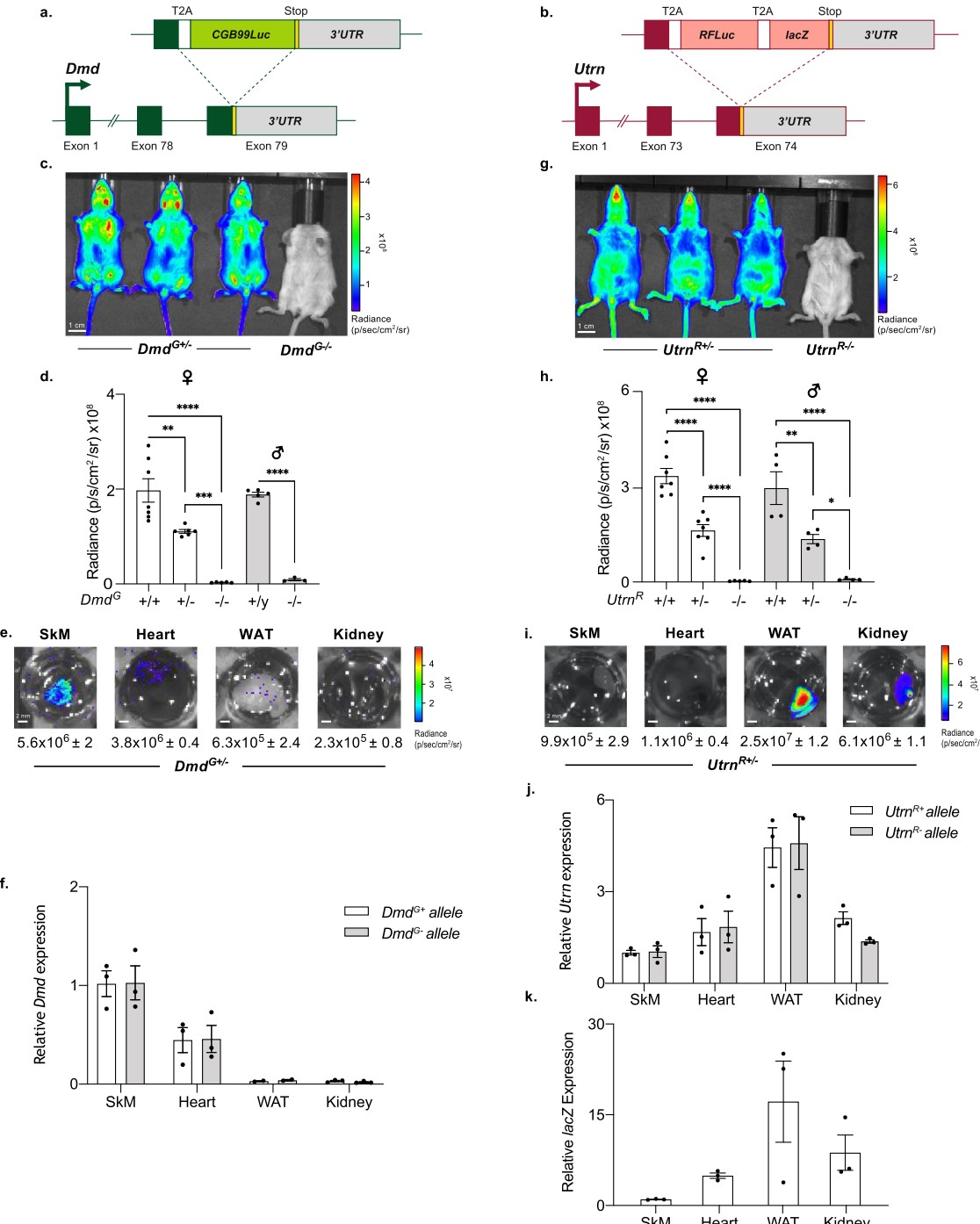

tissues (Fig. S1). These data are consistent with *Utrn* being more widely expressed than *Dmd* in adults[43–46] and reaffirm that luciferase-based targeting had not substantively impaired regulation at the endogenous *Utrn* locus.

**Simultaneous imaging of *Utrn*[R] (RFluc) and *Dmd*[G] (CBG99Luc) expression in vivo.** The targeting strategy we used to generate reporters for *Dmd* and *Utrn* employs different luciferases that respond to the same substrate (D-Luciferin) to produce light with distinct emission spectra; CBG99Luc is predicted to emit light at 540 nm, while RFluc emits light in the red spectrum, at 620 nm[47]. To determine the in vivo emission spectra of our mouse reporters, sequential whole-body images of reporter

mice were taken in the range of 500 nm to 700 nm using discrete 20 nm band pass filters (Fig. 2a). Similar analyses were also performed using isolated ex vivo tissue (Fig. S2a, b). For *Utrn*[R] we observed an emission that peaked at 620 nm as anticipated (Fig. 2a, right). In the case of *Dmd*[G] the spectrum from whole-body scans showed a major peak at 540 nm, with a red-shifted 'shoulder' at around 600 nm (Fig. 2a, left). By analysing the emission spectra of isolated tissues (Fig. S2a), we surmised that this 'shoulder' probably reflects a combination of the intrinsic properties of certain tissues (such as lung), haemoglobin absorption and the reported scattering of light that occurs through deep tissues[42].

To simultaneously image *Dmd* and *Utrn* gene expression in vivo, we crossed the *Dmd*[G] and *Utrn*[R] reporter mice and

**Fig. 1 Imaging *Dmd* and *Utrn* expression in vivo using bioluminescence. a, b** Schematic of the knock in (KI) strategy used to generate $Dmd^G$ (**a**) and $Utrn^R$ (**b**) reporter mouse lines. Reporter genes were inserted into the final exon of the endogenous genes, between the stop codon and the 3'UTR. **a** The *CBG99Luc* gene was knocked into endogenous *Dmd*, separated from the coding region of exon 79 with a T2A site. **b** *LacZ* and *RFluc* genes were knocked into endogenous *Utrn*, separated by T2A sites. **c** Bioluminescent imaging of adult (P28) heterozygous $Dmd^{G+/-}$ female mice showed strong bioluminescent signal corresponding with the location of the musculature and head, while no signal was visible in a wild type (WT) control mouse ($Dmd^{G-/-}$). **d** Quantification of whole-body radiance emitted from $Dmd^{G+/+}$, $Dmd^{G+/-}$, and WT mice showed that bioluminescence emitted from homozygous ($Dmd^{G+/+}$) females was around double that emitted by hemizygous ($Dmd^{G+/-}$) females and equivalent to $Dmd^{G+/y}$ males. Significantly more bioluminescence was detected in all $Dmd^{G+}$ mice than in WT animals. Bar graphs show mean $+/-$ SEM. For each sex, comparisons between genotypes were performed using one-way ANOVA ($p < 0.0001$) with Sidak's multiple comparisons test (adjusted p values: **$p < 0.01$, ***$p < 0.001$, ****$p < 0.0001$). **e** Ex vivo bioluminescent imaging of tissues dissected from heterozygous ($Dmd^{G+/-}$) female mice; highest levels of radiance (blue green) were observed in the skeletal muscle (SkM) and heart, with lower signal in white adipose (WAT) and kidney. Quantification (below) shows mean radiance (p/s/cm²/sr) ($n = 3 +/-$ SD). **f** Allele-specific RT-qPCR, using primers that distinguish mRNA derived from the KI allele ($Dmd^{G+}$, white bars) and the WT allele ($Dmd^{G-}$, grey bars); comparable expression of both alleles was observed in tissues from adult $Dmd^{G+/-}$ female mice. **g** Bioluminescence in adult (P28) heterozygous ($Utrn^{R+/-}$) female mice was detected particularly across the abdomen and head, but not in the WT ($Utrn^{R-/-}$) animal. **h** Quantification of whole-body radiance emitted from $Utrn^{R+/+}$, $Utrn^{R+/-}$ and WT mice. For both sexes, homozygous ($Utrn^{R+/+}$) mice emitted approximately twice the flux of heterozygous ($Utrn^{R+/-}$) mice. Radiance was significantly higher in all $Utrn^{R+}$ mice than WT mice ($Utrn^{R-/-}$). Statistical analysis performed as of **d**. **i** Bioluminescence imaging of ex vivo tissues from $Utrn^{R+/-}$ mice showed strong signal in WAT and kidney, but less in the SkM and heart. Quantification (below) shows mean radiance (p/s/cm²/sr) ($n = 3 +/-$ SD). **j** Allele-specific RT-qPCR on tissues dissected from heterozygous ($Utrn^{R+/-}$) mice showed comparable mRNA expression from the targeted allele ($Utrn^{R+}$, white bars) and the wildtype allele ($Utrn^{R-}$, grey bars). **k** Expression of *lacZ* in tissues from $Utrn^{R+/-}$ mice was quantified by RT-qPCR. **f, j, k** Expression levels were normalised to *18 S* and *Tbp* and are shown relative to the skeletal muscle sample. Bar graphs show mean $+/-$ SEM.

subjected their progeny to consecutive bioluminescent imaging using different emission filters to spectrally separate signal derived from each luciferase (Fig. 2b). Spectral unmixing, using the emission spectra generated from the single reporter strains as a reference, enabled signals derived from the green (CBG99Luc) and red (RFluc) luciferase reporters to be visualised separately within an individual animal. Figure 2b illustrates the success of this approach, showing that mice with compound $Dmd^G$ and $Utrn^R$ genotypes were easily identifiable based on emission. The application of spectral unmixing algorithms, therefore, enabled us to simultaneously visualise $Dmd^G$ expression in the head, jaw and musculature of adult mice, and $Utrn^R$ predominantly in the abdomen (Fig. 2c). Tissues isolated from $Dmd^{G+/+}/Utrn^{R+/+}$ homozygous animals showed the expected distribution of green signal in skeletal muscle and heart, and red signal predominantly in WAT, kidney and to a lesser extent, heart (Fig. 2d). Most importantly, radiance values observed using spectral unmixing (Fig. 2e) were extremely similar to the gene expression profiles for *Dmd* and *Utrn* determined previously by molecular analysis (Fig. 1f, j, respectively).

**Visualising *Utrn* and *Dmd* expression during mouse ontogeny**. Although dystrophin is the predominant paralogue expressed in adult skeletal muscle, utrophin is expressed earlier in development[27,30,48]. To monitor the expression of both paralogues during mouse ontogeny, we performed bioluminescent imaging and molecular analyses using heterozygote embryos generated by crossing $Dmd^G$ and $Utrn^R$ reporter lines. Expression of *Dmd* and *Utrn* was compared between limb tissue from embryos at E11.5, E13.5 and E16.5 stages of gestation and adult skeletal muscle (Fig. 3a, sampled as highlighted in grey). *Dmd* expression was significantly lower in embryonic limb than in adult muscle (Fig. 3a, top), whereas *Utrn* was high in mid-gestation embryos, relative to adult skeletal muscle (Fig. 3a, lower). Bioluminescence images (Fig. 3b) and signal quantification of limb tissue (Fig. 3c) confirmed high level $Utrn^R$ signal in limb tissue between E11.5 and E16.5, which peaked at E13.5. To better discriminate the distribution of *Dmd* and *Utrn* in mid-gestation embryos, we examined single and compound hetero-zygote embryos using spectral unmixing. The emission spectra used for this analysis are shown in Fig. S3a. As shown in Fig. 3d, unmixing revealed red signal exclusively in $Utrn^R$ genotypes, with signal evident in the developing limbs (Fig. 3d). This is consistent

with molecular data showing *Utrn* mRNA transcript detection in tissues from E13.5 embryos (Fig. S3b). To visualise *Utrn*-reporter distribution in embryos at high resolution we used optical pro-jection tomography (OPT) of X-Gal staining of β-galactosidase in $Utrn^{R+/-}$ E13.5 embryos (Fig. 3e and Supplementary Video 1, red). This revealed *Utrn* expression in digits, limbs, tail and spine, results that correspond well with the reported distribution of *Utrn* RNA in E14.5 embryos[49].

**Bioluminescence screen for compounds that enhance *Utrn* expression in adult myoblasts**. Primary and immortalised myoblast cultures were isolated from the skeletal muscle of adult $Utrn^R$ reporters (as illustrated in Fig. 4a) and their capacity to self-renew and differentiate into multinucleated myotubes was confirmed. A myoblast clone U22 A5, which was derived from young a $Utrn^{R+/-}$ adult, was used to establish a bioluminescent platform for $Utrn^R$ upregulation. Briefly, U22 A5 cells were plated at sub-confluence and exposed to either vehicle alone (DMSO) or to chromatin-modifying agents or kinase inhibitors at 100 nM, 1 μM and 10 μM concentrations. $Utrn^R$-derived flux signals were quantified by bioluminescence imaging, comparing levels in vehicle-treated versus test-treated samples. As shown in Fig. 4b, exposure of U22 A5 myoblasts to 5-Azacytidine (5-Aza) or the HDAC inhibitor trichostatin A (TSA) did not increase $Utrn^R$ expression in our hands. However, amongst approximately forty candidates tested, three groups of compounds produced statisti-cally significant increases in $Utrn^R$-derived flux signal in these assays (highlighted in yellow in Fig. 4b); inhibitors of EZH2 (compounds 6 and 7), inhibitors of BRD9 and BRPF1 (com-pounds 15 and 18) and inhibitors of RAF and ERK1/2 (com-pounds 28, 29 and 30). Increased *Utrn* expression in myoblasts treated with EZH2 inhibitors (GSK343, GSK503), bromodomain inhibitors (GSK602, GSK959) or ERK pathway inhibitors (LY3009, Ravox, LY32), but not with 5-Aza or TSA, was inde-pendently confirmed by quantitative analyses of *Utrn*-mRNA transcripts (Fig. 4c). For more detailed information on all the compounds tested see Table S1.

**Conditional PRC2 deletion enhances *Utrn* expression in myoblasts**. EZH2 is a core component of the PRC2 complex, responsible for histone H3K27me3[50]. To confirm a role for EZH2 in regulating *Utrn* expression, we isolated myoblasts in which *Ezh2* could be conditionally deleted. Skeletal muscle from

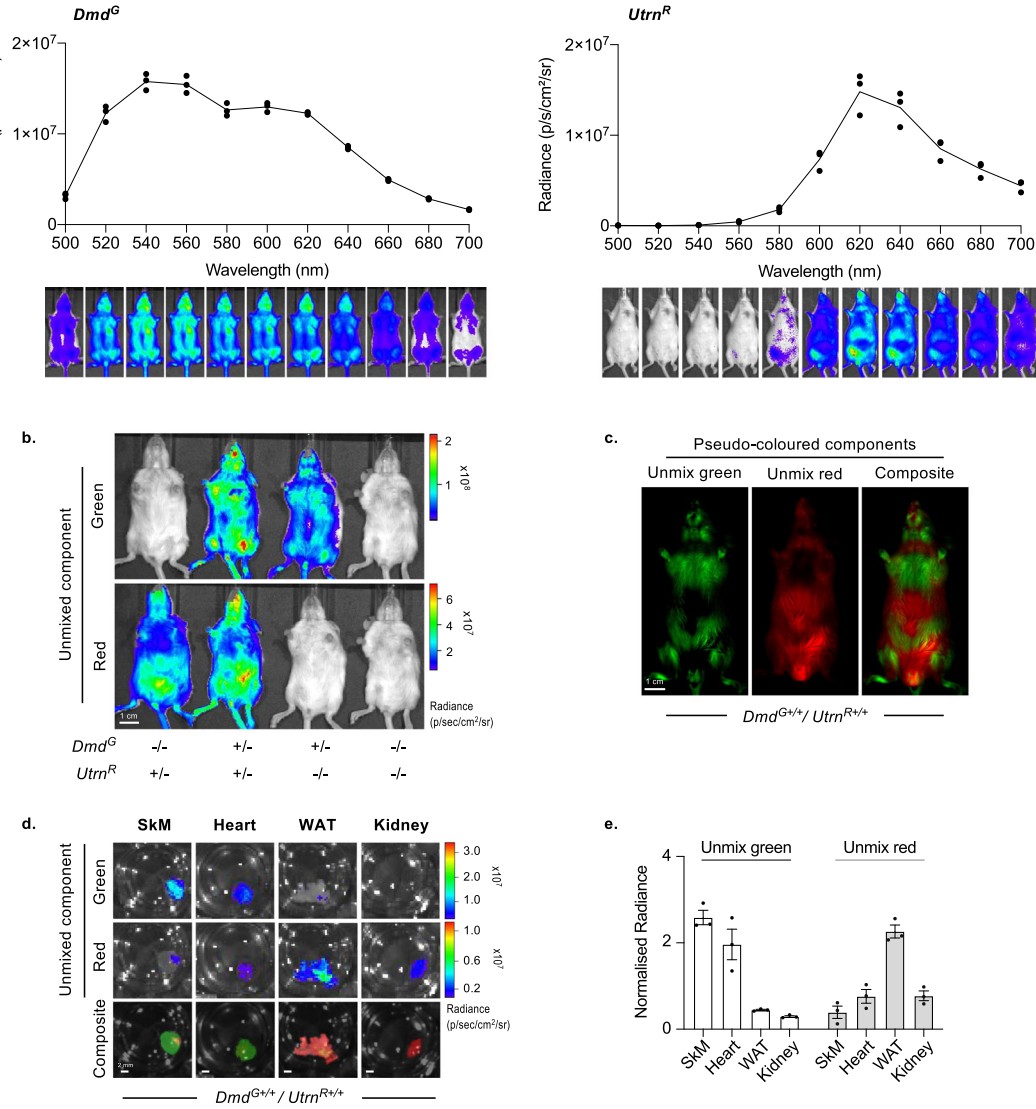

**Fig. 2 Simultaneous visualisation of *Dmd* and *Utrn* in vivo using dual colour bioluminescence imaging. a** Emission spectra were generated from single reporter mouse lines (*Dmd*$^{G+/-}$ left; *Utrn*$^{R+/-}$ right). Bioluminescence images were taken using emission filters across 500–700 nm range with discrete 20 nm band pass filters. Graphs show mean whole-body radiance ($n = 3$) emitted from adult female mice at each wavelength, with representative images below. The peak of *Dmd*$^G$ emission was at 540 nm, with a shoulder at 600 nm, while *Utrn*$^R$ peaked at 620 nm. These spectra were used as references for spectral unmixing analysis in **b**, **c**. **b** Spectral unmixing of *Dmd*$^{G+/-}$, *Utrn*$^{R+/-}$, *Dmd*$^{G+/-}$ / *Utrn*$^{R+/-}$ and WT adult siblings. Unmixing algorithms were able to distinguish bioluminescence emitted from CBG99Luc (green, upper panel) and RFluc (red, lower panel); the green component was visible only in *Dmd*$^G$ $^{+/-}$ and *Dmd*$^{G+/-}$ / *Utrn*$^{R+/-}$ and red only in *Utrn*$^{R+/-}$ and *Dmd*$^{G+/-}$ / *Utrn*$^{R+/-}$ animals. **c** Pseudo coloured images of spectral unmixing of an adult male dual homozygous (*Dmd*$^{G+/+}$/*Utrn*$^{R+/+}$) mouse. Green signal was detected in the skeletal muscle and head, as observed in *Dmd*$^G$ mice, and red was predominant in the abdomen, as observed in *Utrn*$^R$ mice. **d**, **e** Spectral unmixing of tissues dissected from *Dmd*$^{G+/+}$ / *Utrn*$^{R+/+}$ mice. The reference spectra used are shown in Fig. S2. Representative images (**d**) and quantification of the radiance emitted from each component (**e**). SkM and heart emitted green signal, corresponding to *Dmd* expression, while red signal was strongest in WAT, corresponding to *Utrn* expression. Bars show mean $+/-$ SEM.

homozygous mice in which *loxP* sites had been engineered to flank the endogenous *Ezh2* SET encoding domain, and that contain a tamoxifen-inducible CRE cassette at the *Rosa26* locus (Fig. S4a)[51], was used to generate a myoblast cell line (E22). DNA from these cells was analysed before and after 4-OHT addition using pairs of primers that distinguish un-rearranged and excised *Ezh2* alleles (generating fragments of 450 bp or 370 bp, respectively). Kinetic experiments showed that *Ezh2* excision was complete 48–72 h after 4-OHT treatment (Fig. 4d). Consistent with this, *Ezh2* mRNA was no longer detected after 72 h (Fig. 4e and S4b), and we observed a small but significant increase in *Utrn* mRNA transcript levels (Fig. 4e). Western blot analysis confirmed

an increase in utrophin protein abundance, and loss of H3K27me3, in myoblasts in which *Ezh2* had been deleted (compare vehicle and 4-OHT lanes, shown in Fig. 4f and quantified in Fig. S4c). To investigate the possibility that increased utrophin expression might be the direct result of reduced H3K27me3 at the *Utrn* locus we performed H3K27me3 ChIP analysis on U22 A5 myoblasts, examining the *Utrn* promoter, distal enhancer and across the gene body (Fig. S4d). Consistent with ENCODE data for commercial myoblasts we found only very low levels of H3K27me3 across the *Utrn* locus. Treatment of myoblasts with the EZH2 inhibitor GSK503 induced a measurable decline in H3K27me3 at all loci examined, however, this was

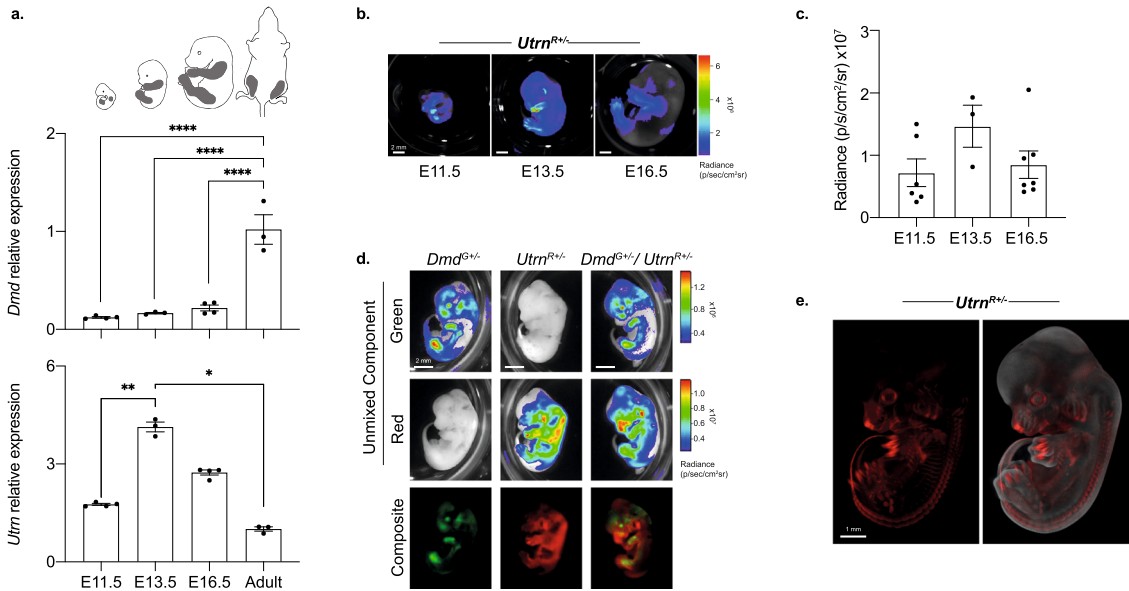

**Fig. 3 *Dmd* and *Utrn* expression during mouse ontogeny. a** RT-qPCR analysis of *Dmd* (upper panel) and *Utrn* (lower panel) mRNA expression in limb tissue during mouse development. Sampled as highlighted in grey. Expression levels were normalised to *18 S* and *Tbp*. Bars show mean +/− SEM; comparison between ages was performed using a one-way ANOVA ($p < 0.0001$ for *Dmd* and $p = 0.008$ for *Utrn*) with Tukey's multiple comparisons test (adjusted *p* values are shown: *$p < 0.05$, ***$p < 0.001$, ****$p < 0.0001$). **b**, **c** Representative bioluminescent images (**b**) and quantification (**c**) of limb radiance from *Utrn*$^{R+/−}$ embryos sampled during embryogenesis. **d** Spectral unmixing of *Dmd*$^{G+/−}$, *Utrn*$^{R+/−}$ and *Dmd*$^{G+/−}$ / *Utrn*$^{R+/−}$ E13.5 embryos. The green component (representing CBG99Luc) was only detected in *Dmd*$^{G+/−}$ and *Dmd*$^{G+/−}$/*Utrn*$^{R+/−}$ embryos, while the red component (representing RFluc) was detected in *Utrn*$^{R+/−}$ and *Dmd*$^{G+/−}$/*Utrn*$^{R+/−}$ embryos. **e** X-Gal staining of *lacZ* expression in *Utrn*$^{R+/−}$ E13.5 embryo, visualised by optical projection tomograph (OPT). LacZ signal is shown in red with total body fluorescence, used to visualise mass, shown in grey. LacZ signal was strongest in the developing digits and spine.

much greater at genes such as *Nlrp6* and *Cntn2* (Fig. S4d), where H3K27me3 is present, than across the *Utrn* locus. Collectively these data provide confirmation that EZH2 negatively impacts *Utrn* expression in myoblasts, but this is unlikely to be driven by changes in H3K27me3 at the *Utrn* locus.

**Inhibitors of ERK1/2 and EZH2 combine to enhance *Utrn* expression**. ERK1/2 inhibitors can also enhance *Utrn* expression in myoblasts. To investigate whether PRC2 and ERK1/2 regulate *Utrn* expression through a shared or entirely separate route, we examined the effects of combining these drugs. To abrogate ERK1/2 signalling, inhibitors were selected that target progressive steps in the pathway (illustrated in Fig. 5a). As illustrated in Fig. 5b, c, U22 A5 myoblasts treated for 24 h with sequential dilutions of the PRC2 inhibitor GSK503 alone, or in combination with the ERK1/2 inhibitors LY32 or Ravox, produced significantly higher *Utrn*$^R$ flux when both pathways were targeted. This observation was verified by quantifying *Utrn* transcripts (Fig. 5d) and extended to show that pairing GSK503 with Raf, MEK or ERK1/2 inhibitors (LY3009, U0126, Ravox and LY32) increased *Utrn* expression by >2-fold, 1.6- fold, >2-fold and 3-fold, respectively.

To determine whether *Utrn* upregulation in response to PRC2 and ERK1/2 inhibitors would also occur in myoblasts that lacked dystrophin, we engineered disease-associated *Dmd* missense mutations into the endogenous locus in U22 A5 myoblasts, using CRISPR/Cas9 (Fig. S5a, b). Two mutant myoblasts lines (clones 2a6 and 3b2) were derived which expressed only low levels of *Dmd* mRNA (Fig. S5c), as anticipated, while *Utrn* expression was unchanged (Fig. S5d). In both *Dmd*-mutant clones we detected significant increases in utrophin gene expression measured by bioluminescence *Utrn*$^R$ flux (Fig. 5e) and transcript analysis (Fig. 5f), 24 h after treating with LY32 and GSK503. These data show that the combined application of PRC2 and ERK1/2

inhibitors also enhances *Utrn* gene expression in dystrophin-deficient myoblasts.

**Utrn upregulation is sustained in myotubes derived from drug-treated myoblasts**. To investigate whether PRC2 and ERK1/2 inhibition enhances utrophin expression in cells at later stages of muscle differentiation, we exposed myoblasts, myotubes and ex vivo isolated myofibres to drugs for 24 h and examined *Utrn* transcript levels. In proliferating myoblast cultures treated with LY32 and GSK503, a 3-fold increase in *Utrn* expression was typically observed (Fig. 6a, upper panel). By allowing these treated myoblasts to differentiate into myotubes in culture by depleting serum (+72 h, 2nd panel) *Utrn* expression remained elevated, relative to untreated matched controls. However, treating differentiated myotubes directly with LY32 and GSK503 did not impact *Utrn* expression (Fig. 6a, 3rd panel). Likewise, *Utrn* expression was not enhanced in primary adult myofibres treated with inhibitors (Fig. 6a, lower panel). These experiments identify proliferating muscle precursors as the target cell type most responsive to inhibitor-driven upregulation of *Utrn*.

One plausible explanation for the utrophin upregulation seen in myoblasts and in treated myoblast-derived myotubes, might be that reduced PRC2 activity and ERK1/2 inhibition enhances myoblast differentiation. A recent study lends some support to this idea[38] as it showed that ERK1/2 inhibition promotes myoblast-to-myotube fusion and growth. To investigate this, we asked whether the expression of myogenesis-associated genes in cultures undergoing serum-induced differentiation was comparable to that seen in myoblasts 24 h after treatment with ERK1/2 inhibitors. As anticipated[52,53], *Pax7* was expressed by proliferating myoblasts and levels declined in cultures exposed to low-serum medium (Fig. 6b, top left). A similar decline in *Pax7* levels was seen in myoblasts exposed to 10 μM LY32 for 24 h (lower

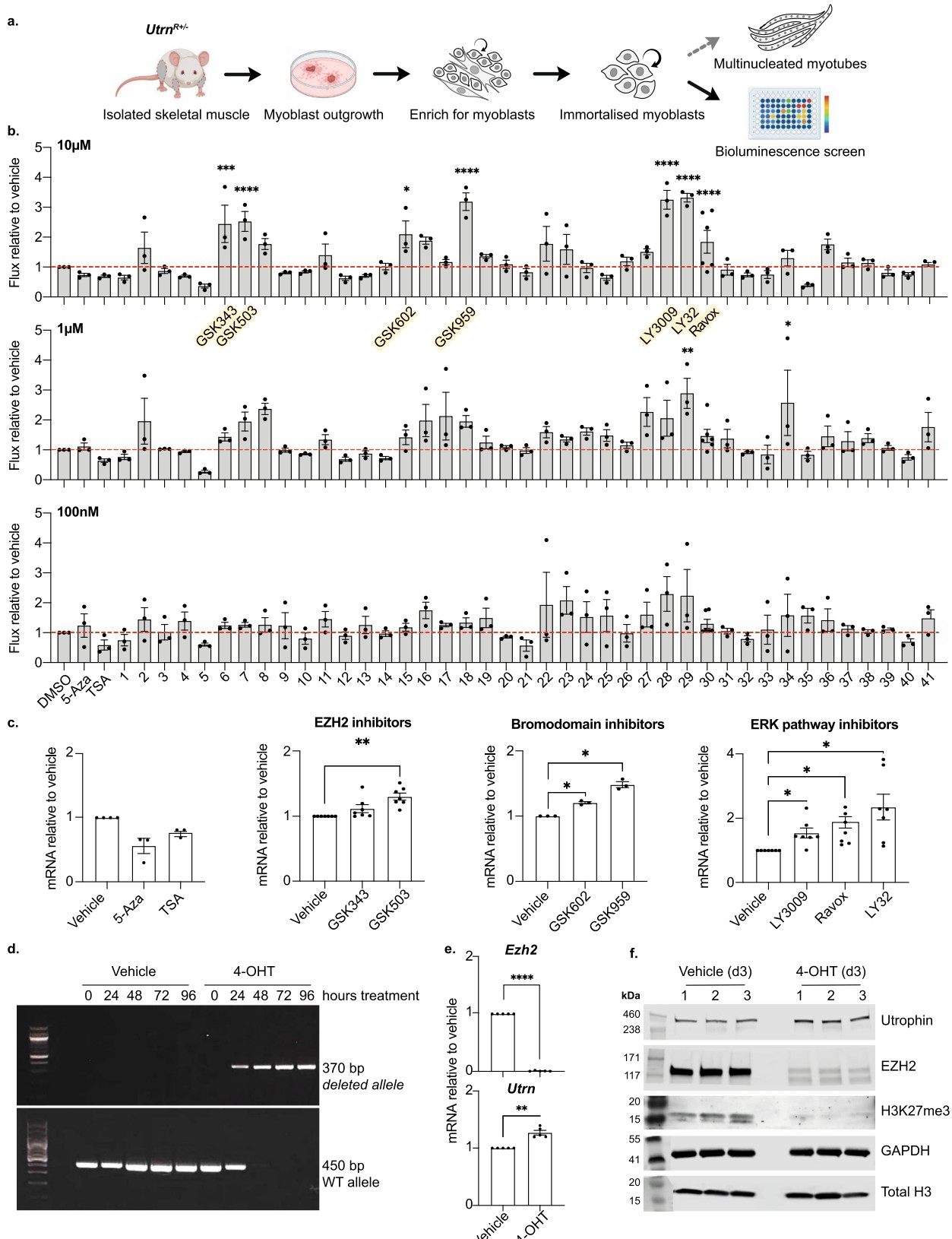

left). Expression of *MyoD*, *Myogenin*, *Mrf4* and *Myh4* genes increased over 3 days of differentiation, as myotubes were formed (Fig. 6b, upper). Treatment of myoblasts with 10 μM LY32 for 24 h resulted in significant increases in expression of each of these genes (Fig. 6b, lower). We observed increased *Dmd* expression later in differentiating cultures (days 2–3) with *Utrn* expression

increasing approximately 3-fold over the differentiation time course. Likewise, a dose dependent increase in both *Utrn* and *Dmd* expression was observed after LY32 treatment. Taken together, our data support the hypothesis that ERK1/2 inhibitors induce the precocious expression of muscle-associated genes that are normally expressed as myoblasts differentiate into myotubes.

**Fig. 4 Generation of a bioluminescent screening platform to identify compounds that increase *Utrn* expression in adult myoblasts. a** Schematic showing the generation of *Utrn*[R+/−] adult myoblasts. Skeletal muscle was isolated from a heterozygous (*Utrn*[R+/−]) female mouse (P22). Muscle tissue was minced and partially digested to encourage outgrowth of myoblasts. Myoblasts were enriched from other cell types by preplating on collagen, immortalised by transfection with SV40 TAg, and clonal lines isolated and used to generate screening platforms. Image created with BioRender.com. **b** Bioluminescent-based screen for compounds that increase *Utrn* expression. *Utrn*[R+/−] myoblasts (U22 A5) were treated with decreasing concentrations of a panel of epigenetic drugs and kinase inhibitors for 24 h (see Table S1 for full list). At 10 μM, EZH2i (GSK503 and GSK343), BRD9i (GSK602), BRPFi (GSK959), RAFi (LY3009) and ERKi (LY32 and Ravox) (highlighted in yellow) treatment increased *Utrn* expression as visualised by an increase in flux (p/s). Bars show flux relative to the vehicle only samples. One-way ANOVAs (10 μM $p < 0.0001$, 1 μM $p < 0.0001$, 100 nM $p = 0.03$) with Dunnett's multiple comparisons test were used to compare vehicle with treated samples. **c** *Utrn* mRNA levels measured by RT-qPCR in U22 A5 myoblasts after treatment with drugs showed that 5-Aza and TSA did not increase *Utrn*, while *Utrn* was increased by the EZH2 inhibitor GSK503, bromodomain inhibitors (GSK602 and GSK959) and ERK pathway inhibitors (LY3009, LY32, Ravox). One-way ANOVAs (in order: $p = 0.07$, $p = 0.003$, $p = 0.008$, $p = 0.013$) with Sidak's multiple comparisons test were used to compare vehicle with treated samples. **d–f** A myoblast cell line (E22) with conditional deletion of EZH2 was generated from a P22 *Ezh2*[flx/flx] male mouse (details are shown in Fig. S4a). **d** PCR showing removal of the WT allele and accumulation of the deleted allele in E22 myoblasts after treatment with 200 nM 4-hydroxytamoxifen (4-OHT) for 24- 96 h. Primer locations are shown in Fig. S4a. **e** RT-qPCR showing *Ezh2* (upper panel) and *Utrn* (lower panel) mRNA expression after 72 h 200 nM 4-OHT treatment in E22 myoblasts. Increased *Utrn* was observed upon *Ezh2* depletion. The comparison between vehicle and 4-OHT treatment was performed using a paired *t*-test. **f** Western blot of E22 myoblasts after 72 h 200 nM 4-OHT treatment for utrophin, EZH2, H3K27me3, GAPDH and total H3. After 4-OHT treatment, EZH2 and H3K27me3 decreased while utrophin increased. Three replicates are shown, quantification is shown in Fig. S4c. The molecular weight (kDa) of the standards is shown alongside. Full blots are shown in Supplementary data Fig. 2. Expression levels (**c**, **e**) were normalised to *18 S* and *Tbp* and are shown relative to the vehicle treated control. Bars represent mean + /− SEM. **b**, **c**, **e** Adjusted *p* values are shown: *$p < 0.05$, **$p < 0.01$, ***$p < 0.001$, ****$p < 0.0001$.

## Discussion

Increased utrophin expression can alleviate the dystrophic phenotype in dystrophin-deficient *mdx* mouse[20–22] and canine DMD models[54]. Utrophin upregulation, therefore, provides a tangible opportunity for therapeutic benefits in DMD. Although dystrophin and utrophin are functionally similar, their expression patterns in normal postnatal life are quite distinct. In adults dystrophin expression is restricted to the brain, skeletal, and cardiac muscles[46], whereas utrophin is widely expressed in a variety of cell and tissue types, including Schwann cells of peripheral nerves, blood vessels, kidney, brain, lungs and adipocytes[43–45]. During ontogeny, utrophin expression at the sarcolemma of embryonic skeletal muscle is progressively replaced by dystrophin[27,29] and eventually becomes restricted to neuromuscular and myotendinous junctions[28,29]. These differences in the temporal distribution of utrophin have encouraged a detailed dissection of *UTRN/Utrn* regulatory elements[3,55–60] and studies to investigate how specialised microRNAs[61,62], artificial transcription factors[63], chromatin modifying drugs[25,64,65] and other agents[33,57,66,67] can upregulate utrophin as a surrogate for dystrophin in postnatal life[20]. The idea of repurposing the expression of a gene that is normally restricted to earlier stages in development, to remedy the loss-of-function of a similar gene expressed in adults, is not without precedent. For example, recent studies showed that targeting the BAF chromatin remodelling complex BCL11A, a repressor complex that silences γ-globin expression in favour of β-globin following birth[68,69], can reactivate γ-globin expression[70]. Although persistent expression of γ-globin into adulthood is rare[71] the application of BCL11A inhibitors to increase γ-globin expression in adult erythroblasts offers promising new treatments for β-hemoglobinopathies[69,70,72].

Despite the development of drugs such as Ezutromid (SMTC1100)[33], Heregulin[34] and specific activators within the AMPK pathway[35,36] that are reported to increase *UTRN/Utrn* transcript levels by 1.4 to 1.8-fold, therapeutic success in DMD remains elusive[73]. Against this background, we developed bespoke preclinical mouse models to enable the endogenous expression of *Dmd* and *Utrn* genes to be simultaneously visualised in vivo and throughout development. Myoblast cell lines were derived from these animals to facilitate the screening of interventions that increase *Utrn* expression and to probe the underlying mechanisms. Bioluminescence screening platforms for utrophin upregulation have been previously described[25,33,61,66],

however, unlike *Utrn*[R] mice and myoblasts, these either lack the full genomic context of *Utrn* or are likely to disrupt the protein structure. Surprisingly, EZH2 inhibitors were shown to increase *Utrn* expression in cultured myoblasts and this finding was confirmed by genetic ablation of *Ezh2* in wildtype myoblasts. HDAC inhibitors that might be predicted to increase *Utrn* expression[25] were not effective in this setting, whilst inhibitors of BRD9 and BRF1 increased *Utrn* expression. As we did not detect appreciable levels of H3K27me3 at the *Utrn* locus in myoblasts it seems unlikely that disruption of PRC2-mediated methylation of lysine 27 at this gene is the root cause. However, recent reports that EZH2 harbours a cryptic transactivation domain that enables interactions with cMyc and p300 and execution of a non-PRC2-dependent activity (i.e., methyltransferase-independent)[74] may be relevant to these observations. Alternatively, canonical EZH2 activity may limit *Utrn* expression indirectly, for example via H3K27me3-mediated repression of *Utrn*-activators or accessory factors.

The potency of ERK1/2 inhibitors to enhance *Utrn* expression in purified myoblasts was also unanticipated, particularly as the prevailing literature suggests that active ERK1/2 drives Heregulin-mediated activity of the utrophin-A promoter[55,57]. A key to understanding this apparent paradox is the recent observation[38] that an evolutionarily conserved signalling cascade, initiated by ERK1/2 inhibition in myoblasts, induces precocious expression of several muscle-associated regulators and enhanced fusion of mononucleated myoblasts in culture. Interestingly, we show here that inhibition of EZH2 and ERK1/2 for 24 h in myoblasts increased *Utrn* expression more than 3-fold, while exposure of cells at later stages of differentiation, such as myotubes and myofibres, did not. We have also shown that drug-exposed myoblasts sustain significantly elevated *Utrn* expression as they subsequently differentiate. Taken together these results raise the intriguing possibility that combined inhibition of EZH2 and ERK1/2 promote the precocious expression of differentiation-associated properties by myoblasts that includes *Utrn* upregulation. As DMD progresses, patients begin to lose regenerative capabilities, therefore an orchestrated action of both enhanced differentiation and increased utrophin expression may be advantageous in a disease setting[25,65,75].

New bioluminescence-based models that allow non-invasive monitoring of gene expression in vivo are beginning to transform our understanding of epigenetics and disease processes[41]. In

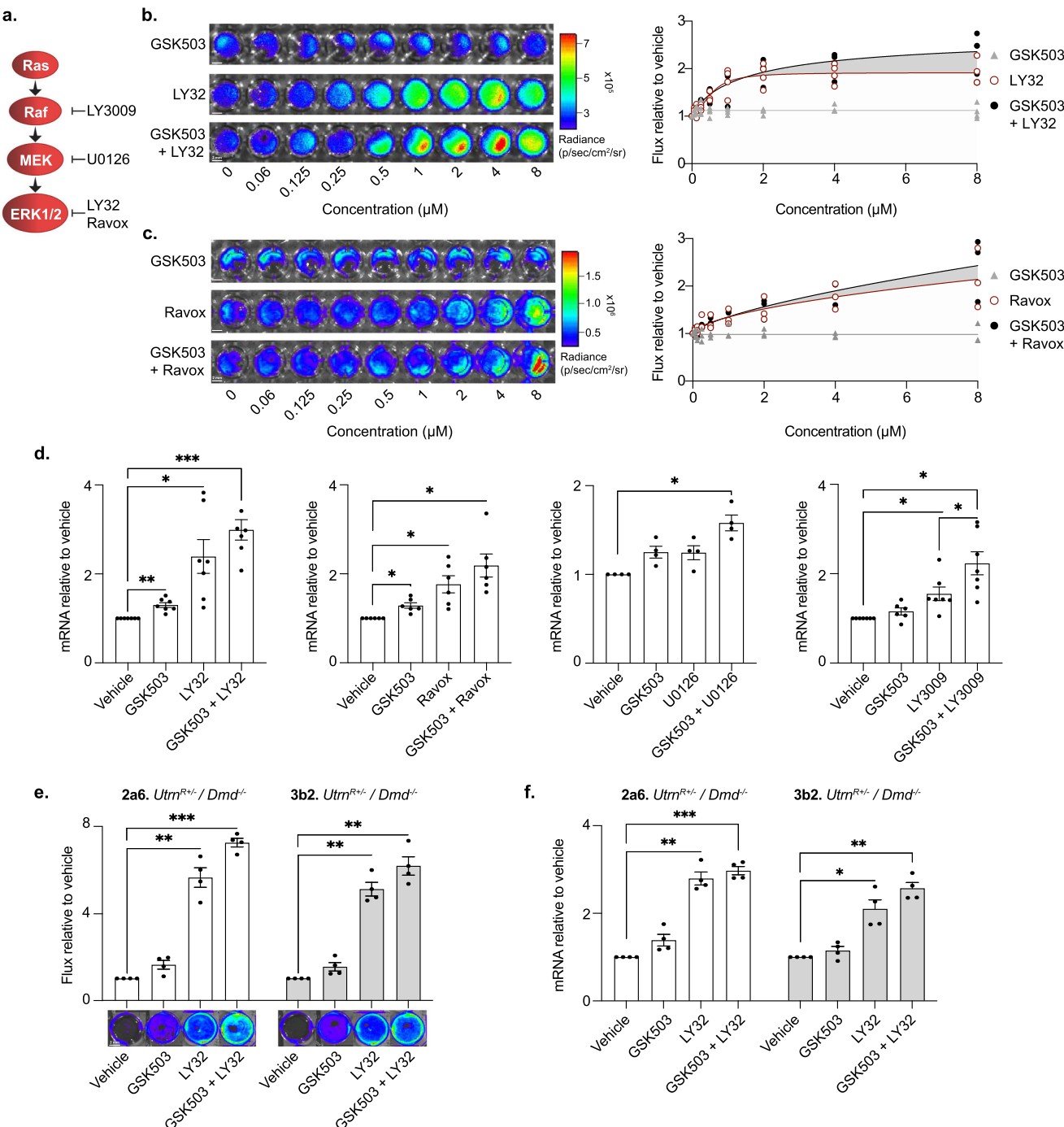

**Fig. 5 ERK inhibitors combined with EZH2 inhibitors increased *Utrn* expression in adult myoblasts. a** Schematic of the ERK signalling pathway and the inhibitors used in this study. **b** Bioluminescence image (left) and quantification of flux (right) of U22 A5 myoblasts after 24 h treatment with EZH2i GSK503 (grey triangles) or ERKi LY32 (red open circles) alone or in combination (black closed circles). LY32 treatment caused a dose-dependent increase in flux which was further amplified by the addition of GSK503 (shaded area). **c** As in **b**, for U22 A5 myoblasts treated with EZH2i GSK503 (grey triangles) and ERKi Ravox (red open circles) either alone or combined (black closed circles). A dose dependent increase in flux was observed after Ravox treatment and this was increased upon the addition of GSK503 (shaded area). **d** RT-qPCR of U22 A5 myoblasts after 24 h treatment with GSK503 and ERK pathway inhibitors (at 10 µM), applied either alone or combined; increased *Utrn* mRNA upon treatment with ERK pathway inhibitors was amplified by the addition of GSK503. One-way ANOVAs (in order: $p = 0.0012$, $p = 0.0047$, $p = 0.0036$, $p = 0.0015$) with Sidak's multiple comparisons test was used to compare between groups. **e**, **f** *Dmd*-deficient U22 A5 myoblast lines 2a6 and 3b2 were generated using CRISPR/Cas9 (details in Fig. S5). *Utrn*[R] bioluminescence (blue green) after treatment with GSK503 or LY32 (10 µM) alone or in combination is shown in **e** and RT-qPCR analysis of *Utrn* mRNA expression is shown in **f**. For each clone, comparisons between treatment groups were performed with one-way ANOVAs (**e**, $p = 0.0009$ both, **f** $p < 0.0001$ and $p = 0.0023$) with Dunnett's multiple comparisons test. Expression levels were normalised to *18 S* and *Tbp* and shown relative to the vehicle treated control (**d**, **f**). Bars show mean $+/-$ SEM with adjusted *p* values: *$p < 0.05$, **$p < 0.01$, ***$p < 0.001$) (**d**–**f**).

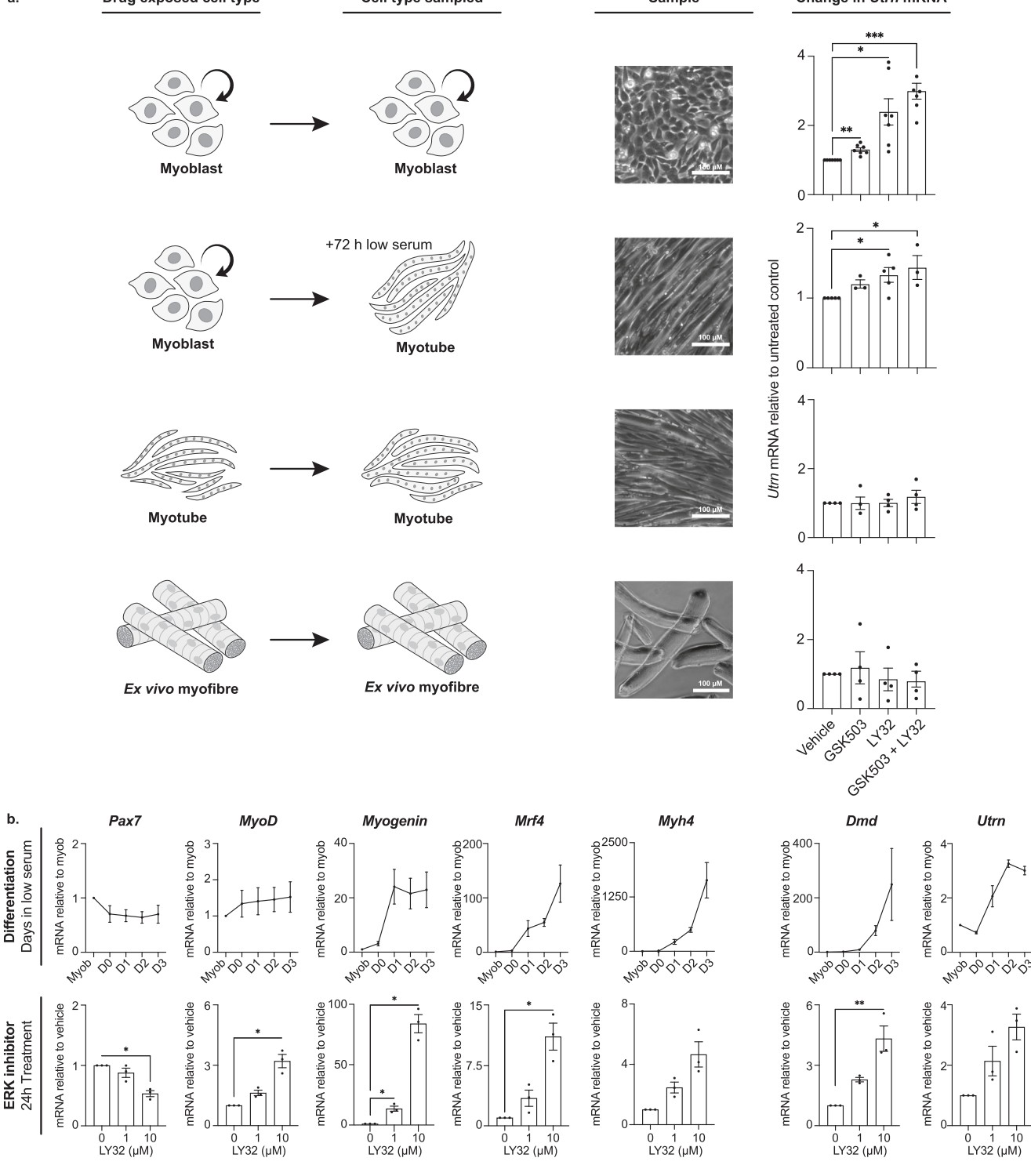

**Fig. 6 *Utrn*-induction by ERK1/2 and EZH2 inhibitors is selective for myoblast stages and is sustained following differentiation. a** Schematic showing different types of muscle cells that were isolated and cultured (left) before being exposed to GSK503 or LY32 alone (10 μM) or in combination. Expression of *Utrn* mRNA was quantified by RT-qPCR after 24 h (right) and a representative phase contrast image of the cells at the time of sampling is shown (middle). As shown in Fig. 5d treatment of proliferating U22 A5 myoblasts with these drugs induced *Utrn* expression (upper panel), which was maintained following differentiation (72 h serum depletion, 2nd panel), while cultured myotubes (3rd panel) or ex vivo myofibres (lower panel) were unresponsive. *Utrn* mRNA levels were normalised to *18 S* and *Tbp*, and expressed relative to vehicle treated samples. One-way ANOVAs ($p = 0.0012$, $p = 0.028$, $p = 0.75$, $p = 0.8$) with Dunnett's multiple comparisons test was used to compare between groups. **b** RT-qPCR analysis of the temporal expression of genes associated with specific stages of myoblast differentiation in U22 A5 myoblasts upon serum depletion (upper panels), and corresponding gene expression responses to 0, 1 μM and 10 μM ERK1/2 inhibitor LY32 (lower panels). Serum-deprived samples (upper) are normalised to *Csnk2a2* while drug-treated samples (lower) were normalised to *18 S* and *Tbp*. One-way ANOVAs with Dunnett's multiple comparisons test were used to compare vehicle and treated groups. Graphs show mean +/− SEM with adjusted *p*-values: *$p < 0.05$, **$p < 0.01$, ***$p < 0.001$.

addition, bioluminescence-based cell reporters for *Utrn* are being successfully applied to screen for small molecule modulators of utrophin[66]. The capacity to longitudinally view the expression of multiple genes in concert[42,76] offers new opportunities to assess the timing and co-ordinate responses of gene regulatory circuits to drugs and environmental challenges. Bioluminescence imaging has been used to successfully monitor dystrophin expression and evaluate DMD correction therapies in mice[77]. Here, we generated bespoke *Dmd^G/Utrn^R* dual reporter mice to simultaneously examine *Dmd* and *Utrn* expression in vivo and as preclinical tools to generate a screening platform to uncover agents capable of enhancing utrophin expression in myoblasts and differentiating muscle. We provide proof of principle that this approach can robustly identify new candidates for future therapeutic evaluation. In addition, our dual reporter lines provide a platform to assess whether *Utrn* upregulating therapies promote co-localisation of *Utrn* with *Dmd* in muscle tissue and offer unprecedented access to the developmental stages in which *Utrn* expression becomes 'down-graded' from the sarcolemma in embryos to the neuro-muscular and myotendinous junctions postnatally[27,78].

## Methods

**Animal maintenance**. All animal procedures were performed in accordance with the British Home Office Animal (Scientific Procedures) Act 1986 and the GSK Policy on the Care, Welfare and Treatment of Animals. The mouse work was approved by the Imperial College AWERB committee and performed under a UK Home Office Project Licence and Personal Licences.

Mice were housed in a pathogen free facility at temperatures of $21+/-2\,°C$; 45–65% humidity; 12-h light-dark cycle; with water and RM3 diet *ad libitum*. Tissues, wood blocks, and tunnels were used to enrich the environment. Experiments on adult mice were performed on animals between 3–10 weeks old.

For timed mating, an adult male was set up with 2 adult females and morning plug checking was performed. The females were separated from males upon discovery of vaginal plugs, at which point they were considered to be at E0.5 stage of pregnancy.

**Animal lines**. Dmd-CBG99Luc (referred to as *Dmd^G*) mESCs and mouse line were generated by Taconic Biosciences; the Click Beetle Green 99 luciferase gene (*CBG99Luc*) was knocked into the 3′UTR of the endogenous *Dmd* gene, separated by a T2A sequence (see Fig. 1a for details). Utrn-RFluc-lacZ (referred to as *Utrn^R*) mESCs and mouse line were engineered by OzGene; a Red Firefly Luciferase (*RFluc*) and the *lacZ* gene were inserted into the 3′UTR of endogenous *Utrn*, separated by T2A sequences (Fig. 1b). Both mouse lines were crossed into a C57BL6/Albino background to improve resolution for in vivo bioluminescent imaging.

The *Ezh2^flx/flx* mouse strain was provided by A. Tarakhovsky (The Rockefeller University)[51] and crossed with *Rosa26^ERt2-Cre/ERt2-Cre* mice to generate tamoxifen inducible *Ezh2* knock out animals (*Ezh2^flx/flx* / *Rosa26^ERt2-Cre/ERt2-Cre* (referred to as *Ezh2^flx/flx*)).

**Genotyping**. Genomic DNA was isolated from ear biopsies of 3–4-week-old mice. Ear samples were incubated in 100 μl lysis buffer (100 mM NaCl, 10 mM Tris pH 7.5, 10 mM EDTA, 0.5% N-Lauroylsarcosine sodium salt) overnight at 55 °C. Samples were diluted 1:2 in dH₂0, and centrifuged for 5 min at top speed. 1 μl of the supernatant was used in PCR reactions.

Genotyping primers
*Dmd^G* WT/KI allele Fwd: ACCAGCTTGAAATTTGCCC
*Dmd^G* WT/KI allele Rev: TTGCCCACAAGCACTTGAC
*Utrn^R* KI allele Fwd: GTTTCCCTCTTGCAGCTCAA
*Utrn^R* KI allele Rev: ATCGGGTAGAATGGCAGTGG
*Utrn^R* WT allele Fwd: GGACGTGATGGAGCAGATCA
*Utrn^R* WT allele Rev: GAGCTTTGGGGGTTGAATGGG
*Ezh2* Lox P Fwd: TTATTCATAGAGCCACCTGG
*Ezh2* Lox P Rev: CTGCTCTGAATGGCAACTCC
*Rosa26* Fwd: AAGGGAGCTGCAGTGGAGT
*Rosa26* Rev: GTCCCTCCAATTTTACACC
*CreER^t2* Fwd: ACGAGTGATGAGGTTCGCAA
*CreER^t2* Rev: AGCGTTTTCGTTCTGCCAAT

**In vivo and ex vivo bioluminescent imaging**. For in vivo bioluminescent imaging adult mice were intraperitoneal (IP) injected with 0.15 mg/g D-Luciferin (Perkin Elmer), dissolved in dH₂O. Mice were left conscious for 3 min to allow the D-Luciferin to circulate, then anesthetized with isoflurane. 10 min post injection mice were imaged using the IVIS Spectrum (Perkin Elmer) and Living Image software (version 4.3.1). Adult mice were imaged using field of view (FOV) C or D,

bin 1 with 15 s exposure time using a stage temperature of 37°C. Dissected tissues were incubated in 150 μg/ml D-Luciferin in PBS for 10 min prior to imaging.

For imaging of embryos, pregnant females were IP injected with 0.15 mg/g D-Luciferin, and embryos dissected 12 min post injection. Dissected embryos were placed in 24 well plates and incubated in 150 μg/ml D-Luciferin in PBS for 10 min prior to imaging on the IVIS Spectrum. Embryos were imaged using FOV A, bin 4 and 60 s exposure.

All single reporter imaging was performed using the 'open' emission filter. Image analysis was performed using the Living Image software (version 4.5.2) (Perkin Elmer). For bioluminescence quantification, regions of interest (ROI) were drawn around animals, embryos or tissues to calculate flux (p/s) and average radiance (p/s/cm²/sr) within the region.

**Dual colour bioluminescent imaging and spectral unmixing**. Consecutive images of *Dmd^G* and *Utrn^R* single reporter mice, embryos and tissues were taken on the IVIS Spectrum using emission filters with wavelengths ranging from 500 to 700 nm with 20 nm band pass. Quantification of the flux generated from each emission filter was used to produce an emission spectrum for each luciferase. These spectra were saved as reference spectra on the Living Image software. Dual reporter (*Dmd^G/Utrn^R*) mice, embryos and tissues were similarly imaged using emission filters between 500 to 700 nm. The spectral unmixing algorithm on the Living Image software was used to separate the signals from the two luciferases. The manual spectral unmixing setting was used, with the emission spectra generated from the single reporter mice inputted as reference spectra. This separates the CBG99Luc signal from the RFluc signal.

**X-Gal staining**. Tissues from *Utrn^R* and WT mice were incubated in cold lacZ fixative (2% formaldehyde, 0.2% glutaraldehyde, 0.02% Nonidet P-40, 1 mM MgCl₂, 0.1 mg/ml Sodium Deoxycholate in PBS) for 1 h at 4 °C. Samples were incubated in 30 % sucrose for 24 h, embedded in O.C.T compound and stored at −80 °C. Frozen tissue was sectioned to 20 μm thickness using a cryostat and collected onto Super-frost slides. Defrosted slides were stained with lacZ stain (0.5 mg/ml X-Gal, 4 mM Potassium Ferrocyanide, 4 mM Potassium Ferricyanide, 1 mM MgCl₂, 0.02% Non-idet P-40 in PBS) at 37 °C for 5 h. After staining, tissue was fixed in 4% PFA for 10 min then washed (1 × 5 min PBS, 1 × 10 min PBS, 2 × 5 min dH₂O). Counter staining was performed by 2 min incubation with Nuclear Fast Red solution (Sigma) followed by two 5 min washes in dH₂O. Slides were mounted using Prolong Gold Antifade mountant (Thermo Fisher Scientific) and imaged on an Axio Scan.Z1 Slide Scanner (Zeiss).

**Optical projection tomography (OPT)**. Dissected embryos were incubated in cold lacZ fixative for 1 h at 4 °C. Embryos were washed in PBS prior to incubation with lacZ stain (0.4 mg/ml X-Gal, 4 mM Potassium Ferrocyanide, 4 mM Potassium Ferricyanide, 1 mM MgCl₂, 0.02% Nonidet P-40 in PBS) with rocking for 24 h at 4 °C. Embryos were washed twice in PBS, once in 70% ethanol and rehydrated in dH₂O for 24 h at 4 °C. Embryos were mounted in cylinders of 2% low melting point agarose. The mounted samples were dehydrated through graded methanol solutions and maintained in 100% methanol prior to clearing. They were subsequently immersed overnight in an optical clearing solution, BABB (1:2 Benzyl benzoate: Benzyl alcohol, Sigma Aldrich). Following clearing, optical projection tomography (OPT)[79] was performed[39]. Briefly, the cleared samples were suspended from a rotation stage (T-NM17A200, Zaber Technologies Inc) in a cuvette filled with BABB and imaged using a 1× telecentric lens (58–430, Edmund Optics Ltd) with images recorded using a CCD camera operated at 2 × 2-pixel binning (Zyla 5.5, Andor Technology Ltd). To measure the distribution of lacZ staining, transmitted light images were acquired every 0.9° over a full 360° sample rotation through a 708 ± 37 nm band-pass filter (FF01-708/75-25, Laser 2000 UK Ltd). Reconstruction was performed using a filtered back-projection algorithm[80] to reproduce a 3-D label distribution (displayed in red). In addition to the transmitted light data, fluorescence OPT acquisitions using a 465 nm excitation laser source (Tri-Line laser bank, Cairn Research Ltd) imaged through the same 708 nm filter were performed, reconstructing the whole sample volume (shown in greyscale).

**Generation of adult myoblast cell lines**. Myoblast cell lines were generated from 3–4-week-old mice. The U22 A5 *Utrn^R+/−* line and the E22 *Ezh2^flx/flx* myoblast line used in this study were generated from a 22-day-old heterozygous *Utrn^R+/−* female and a 22-day-old homozygous *Ezh2^flx/flx* male, respectively. The protocol used to generate myoblast lines was adapted from Shahini et al., 2018[81]. Skeletal muscle was isolated from the hind limbs of the mouse, minced using scissors, and incubated with 1 ml enzymatic solution (0.25 mg/ml Collagenase type II solution (Sigma), 1.5 U/ml Collagenase D solution (Roche), 2.5 U/ml Dispase II solution (Sigma) and 2.5 mM CaCl₂, in PBS) at 37 °C for 30 min. The partially digested tissue was plated onto 0.9 mg/ml Matrigel-coated plates in proliferation medium (PM) (high glucose DMEM supplemented with 10% horse serum (Gibco), 20% heat inactivated FBS (Fisher Scientific), 5 ng/ml basic fibroblast growth factor (R&D Systems), 0.5% chicken embryo extract (Fisher Scientific) and 1% antibiotic-antimycotic (Gibco)). After 3–5 days, the tissue starts producing myoblasts. These were harvested by washing off the pieces of tissue then detaching the outgrowth population using trypsin. Cells were preplated on 0.1 mg/ml collagen (Corning)

coated plates for 30 min, then the cells in the supernatant plated at 3000 cells/cm$^2$ on 0.09 mg/ml Matrigel-coated plates in PM. Preplating on collagen was performed for the first four passages to enrich for myoblasts.

Primary myoblasts were immortalised using the simian virus 40 (SV40) large T-antigen (TAg), isolated from the TAg producing cell line Q2TSA583[82]. Myoblasts were exposed to conditioned media from Q2TSA583 cells in the presence of 8 μg/ml polybrene (Sigma) for 72 h, following which selection was performed (96 h 0.5 mg/ml Geneticin (ThermoFisher)) and single cell colonies isolated. The presence of TAg in the clones was verified by PCR. Myoblast cell lines were maintained in proliferation medium on 0.09 mg/ml Matrigel-coated plates.

TAg Primers
TAg Fwd: AGCATTATGCAAATGCTGC
TAg Rev: AGCCATCCATTCTTCTATGT

To promote differentiation and the formation of multinucleated myotubes, myoblasts were grown to 90% confluency on 0.09 mg/ml Matrigel-coated plates. Proliferation medium (PM) was replaced with differentiation medium (DM) (high glucose DMEM, 2% horse serum and 1% antibiotic-antimitotic) and cells were allowed to differentiate for 72 h.

**Bioluminescent drug screen**. Drugs used in this study were purchased from Sigma or SelleckChem, or gifted from GSK. All drugs were dissolved in DMSO and all treatments were performed for 24 h.

For the bioluminescent drug screen, U22 A5 myoblasts were plated on 96-well plates 24 h prior to treatment. Drugs were added to a final concentration of 100 nM, 1 μM or 10 μM and incubated for 24 h. An untreated and vehicle (DMSO) treated well was included in each experiment. Following treatment, 150 μg/ml D-Luciferin was added and the plate imaged by the IVIS Spectrum using FOV C, Bin 4 and 60 s exposure. Images were analysed using the Living Image Software with ROI drawn around each individual well and flux calculated. Total flux from treated samples was normalised against the vehicle treated wells.

Further myoblast treatment experiments were performed as described above using 12- or 6- well plates. For treatment of myotubes, myoblasts were grown to confluency in PM, differentiated in DM for 48 h, prior to 24 h drug treatment. Myoblast pretreatment experiments involved first treating the myoblasts in PM for 24 h, then changing to DM to differentiate for 72 h prior to sampling.

**4-Hydroxytamoxifen (4-OHT) treatment**. 4-OHT resuspended in 100% ethanol (EtOH) was added to E22 Ezh2$^{flx/flx}$ cells for 24 to 96 h at a concentration of 200 nM to mediate site- specific excision of the loxP sites and deletion of the Ezh2 SET domain (details shown in Fig. S4a). Primers specific for the excised allele or the WT allele were used to confirm efficacy of the knock out. For follow on experiments involving E22 cells, 4-OHT treatment was performed for 72 h.

Primers to verify Ezh2 KO
Ezh2$^{flx/flx}$ excised allele Fwd: ACGAAACAGCTCCAGATTCAGGG
Ezh2$^{flx/flx}$ excised allele Rev: CTGCTCTGAATGGCAACTCC
Ezh2$^{flx/flx}$ WT allele Fwd: TTATTCATAGAGCCACCTGG
Ezh2$^{flx/flx}$ WT allele Rev: CTGCTCTGAATGGCAACTCC

**CRISPR/Cas9-mediated gene editing**. sgRNAs targeting the mouse Dmd gene were designed using the Benchling CRISPR design tool (https://www.benchling.com). The corresponding oligonucleotides were cloned into human codon optimised SpCas9 and chimeric guide RNA expressing plasmid, pX330-U6-Chimeric_BB-CBh-hSpCas9 (a gift from Feng Zhang, Addgene plasmid #42230[83]). For gene editing, U22 A5 myoblasts were co-transfected with the constructed CRISPR/Cas9-sgRNA plasmid targeting mouse Dmd and an mCherry expressing plasmid using Lipofectamine 3000 (Invitrogen). Two days after transfection, mCherry expressing cells were individually sorted into wells of a 96-well plate using fluorescence-activated cell sorting, were expanded and genotyped by sanger sequencing to identify mutations.

sgRNA sequences
sgDmd 2: AATGAATGACATGCGCCCAA
sgDmd 3: GTTTTAGAATTCCCTGGCGC

**Myofibre Isolation**. The myofibre isolation protocol was adapted from Ravenscroft et al., 2007 and Shefer et al., 2005[84,85]. Flexor digitorum brevis (FDB) muscle was dissected from adult WT mice in warm PRS (138 mM NaCl, 2.7 mM KCl, 1.8 mM CaCl$_2$, 1.06 mM MgCl$_2$, 12.4 mM HEPES, 5.6 mM Glucose, pH 7.3) and incubated with digestion solution (2 mg/ml collagenase, 10% FBS in PRS) for 1.5 h at 37 °C. Digestion solution was replaced with proliferation media (DMEM supplemented with 10% FBS, 1% Pen-Strep (Gibco) and 4 mM L-glutamine (Gibco)) and incubated for an additional 30 min. Myofibres were released from partially digested tissue by first removing tendons and connective tissue, then pipetting muscle tissue through Pasteur pipettes of decreasing width to separate the fibres. Density sedimentation was used to separate undamaged myofibres from single cells and damaged fibres by passing the suspension through three tubes of 10 ml proliferation medium. Myofibres were plated on 20 μg/ml laminin coated plates in maintenance media (DMEM containing 20% controlled serum replacement-2 (Sigma), 4 mM L-glutamine and 1% Pen-Strep).

**Reverse transcription quantitative real-time PCR (RT-qPCR)**. RNA was extracted from cultured cells and tissue samples using the RNeasy Mini kit (Qiagen). Tissue samples were first lysed in RLT buffer on the TissueLyserII (Qiagen) using 5 mm stainless steel beads (Qiagen) for 4 min at 24,000 rpm. Skeletal muscle and heart samples were incubated with 10 μg/ml Proteinase K at 55 °C for 1 h. All tissue samples were then centrifuged at top speed for 3 min and RNA was purified from the supernatant. Cell pellets were directly lysed in RLT buffer. The RNeasy Mini Kit was used according to the manufacturer's instructions with on column DNAse treatment (Qiagen). cDNA was generated using SuperScript III reverse transcriptase (Thermo Fisher) according to the manufacturer's instructions, with 10 μM random primers.

RT-qPCR was performed using the QuantiTect SYBR Green PCR mix (Qiagen) with 10 nM forward and reverse primers. All samples were run in technical triplicate and the mean used for further analysis. Efficiency was calculated for all primer pairs using the formula $E = 10^{-1/Slope}$ and relative RNA levels were determined using the formula $EI^{control\ Ct\ -\ sample\ Ct} / EH^{control\ Ct\ -\ sample\ Ct}$, where EI = efficiency of primers of interest and EH = efficiency of housekeeping primers. Control samples were either vehicle treated samples or undifferentiated myoblasts for differentiation experiments, details are provided in the figure legend. The mean of 18 S and Tbp was used for normalisation for tissue samples and drug treated cells, while Csnk2a2 was used for myoblast differentiation experiments.

Allele-Specific RT-qPCR primers
Dmd$^{G+}$ allele Fwd: AATGGAGGGCAGAGGAAGT
Dmd$^{G+}$ allele Rev: CGGCAGTCAAATCCTCCAAA
Dmd$^{G-}$ allele Fwd: CGGAAAGCCAATGAGAGAG
Dmd$^{G-}$ allele Rev: TGCCCAAATCATCTGCCATG
Utrn$^{R+}$ allele Fwd: GTTTCCCTCTTGCAGCTCAA
Utrn$^{R+}$ allele Rev: ATCGGGTAGAATGGCAGTGG
Utrn$^{R-}$ allele Fwd: GGACGTGATGGAGCAGATCA
Utrn$^{R-}$ allele Fwd: GAGCTTTGGGGTTGAATGGG

RT-qPCR primers
Dmd Fwd: GGCAGATGATTTGGGCAGAG
Dmd Rev: CATGCGGGAATCAGGAGTTC
Utrn Fwd: ACTATGACCCCTCCCAGTCC
Utrn Rev: ATCCTCCACGCTTCCTGTTG
lacZ Fwd: GGTGACCACAGGATAGGCAT
lacZ Rev: TGCCCACAGTACTCAAGGTT
18 S Fwd: GTAACCCGTTGAACCCCATT
18 S Rev: CCATCCAATCGGTAGTAGCG
Tbp Fwd: AGCTCTGGAATTGTACCGCA
Tbp Rev: TGACTGCAGCAAATCGCTTG
Csnk2a2 Fwd: CCAGTTGACAGTGCCCTTTC
Csnk2a2 Rev: TTCTCGATGGCCTCAAACCT
Ezh2 Fwd: GGTGACCACAGGATAGGCAT
Ezh2 Rev: TGCCCACAGTACTCAAGGTT
Pax7 Fwd: TGCCCTCAGTGAGTTCGATT
Pax7 Rev: GATGCCATCGATGCTGTGTT
MyoD1 Fwd: ATGATGACCCGTGTTTCGA
MyoD1 Rev: CACCGCAGTAGGGAAGTGT
Myogenin Fwd: GAGACATCCCCCTATTTCTACGA
Myogenin Rev: GCTCAGTCCGCTCATAGCC
Mrf4 Fwd: CTGAAGCGTCGGACTGTGG
Mrf4 Rev: ATCCGCACCCTCAAGAATT
Myh4 Fwd: GCCCAGAACAAGCCTTTTGA
Myh4 Rev: ACTTGGGAGGGTTCATGGAG

**Chromatin immunoprecipitation (ChIP)**. Chromatin immunoprecipitation was adapted from Nelson et al., 2008[86]. U22 A5 myoblast cells were fixed with 1% formaldehyde (Thermo Scientific) at room temperature (RT) for 10 min, followed by quenching with 125 mM glycine for 5 min. Myoblasts were washed in cold PBS, harvested by cell scraping and centrifuged at 376 rcf for 5 min. The pellet was resuspended in 900 μl SDS lysis buffer (1% SDS, 50 mM Tris HCl pH 8.1 and 10 mM EDTA in nuclease free water (NF-H$_2$O)) supplemented with 1x cOmplete Protease Inhibitor Cocktail (Roche), and incubated for 30 min on ice. Sonication was performed on a Bioruptor Plus (Diagenode) at high power for 20 cycles of 30 s on/off.

Chromatin was incubated with 20 μl prewashed Protein A Sepharose beads (Sigma) for 1 h at 4 °C to pre-clear the sample. Antibodies (H3K27me3, Cell Signalling 9733 S, 5 μg/immunoprecipitation (IP); total H3, Abcam ab1791, 3 μg/IP) were incubated with 10 μg of precleared chromatin in ChIP dilution buffer (167 mM NaCl, 16.7 mM Tris pH 8.1, 1.1% Triton X-100, 1.2 mM EDTA and 0.01% SDS in NF-H$_2$O) overnight at 4 °C.

The following day, 50 μl prewashed Protein A Sepharose beads were added to the antibody-bound chromatin and incubated at 4 °C for 4 h with rotation. Chromatin- bound beads were sequentially washed in Low Salt Immune Complex Wash Buffer (150 mM NaCl, 20 mM Tris pH 8.1, 0.1% SDS, 1% Triton X-100 and 2 mM EDTA in NF-H$_2$O), High Salt Immune Complex Wash Buffer (500 mM NaCl, 20 mM Tris pH 8.1, 0.1% SDS and 2 mM EDTA in NF-H$_2$O), LiCl Immune Complex Wash Buffer (0.25 mM LiCl, 10 mM Tris pH 8.1, 1% NP-40 and 1 mM EDTA in NF-H$_2$O) and TE buffer. Each wash consisted of 4 min rotating at RT followed by 1 min centrifugation at 1150 rcf.

100 µl Chelex-100 (Sigma) was used to isolate the enriched DNA: samples were boiled for 10 min, incubated with Proteinase K for 1 h at 55 ℃ and then boiled for a further 10 min. The sample was centrifuged at 13,523 rcf for 1 min to separate the Sepharose beads and resin from the DNA. The supernatant, containing the DNA, was retained for use in ChIP-qPCR experiments. The same protocol was used to extract DNA from 1 µg of chromatin as a 10% input sample.

ChIP-qPCR was performed using the QuantiTect SYBR Green PCR mix. Efficiency was determined for all primers used and percentage input calculated using the following formula $(E^{adjusted\ input\ Ct\ -\ sample\ Ct})$ x100, where E = primer efficiency, and adjusted input = Ct of 10% input − 3.32.

ChIP qPCR Primers:

Controls

*Cntn2* Fwd: ACACTGGTAACCTGCAATGG
*Cntn2* Rev: TTCAGTCTTCCCGAGCATGT
*Nlrp6* Fwd: TTTGGAGGTTCAGGGACAGG
*Nlrp6* Rev: GCTCCTCCAGTGTAGCCATA
*Gapdh* Fwd: GCCAGGAAGACGCTTGAAAA
*Gapdh* Rev: GGGTCCCAGCTTAGGTTCAT

*Utrn* locus

*Ex63* Fwd: GCCAGAAATCAGTGTGAAGGA
*Ex63* Rev: GCCTGATGTTTTGCAGTCTCA
*Ex46* Fwd: AATACCAGCCAGTCAGCCAG
*Ex46* Rev: GGTGGTGCTAGTATCATCTGC
*Ex26* Fwd: ATTCCGACGTCAACTCCTGG
*Ex26* Rev: AAGAGGAGGTGCTGCAGAAG
*In11* Fwd: GTGCAAAATGAGGAAAAGGCTG
*In11* Rev: AGAGGGAAGTGAAGGTGTCC
*Ex6* Fwd: TGGAGCACGGCGTTGAAC
*Ex6* Rev: GCGAGAAGATCCTGCTGAGC
*DUE 1* Fwd: CCTCCCTTTGAATGGCACCT
*DUE 1* Rev: TCCAAATTGATCTGCCAGCT
*DUE 2* Fwd: CCAGCCAGAGGAATGTGATT
*DUE 2* Rev: TGTTATAACTGTGCCCTCCCT
*DUE 3* Fwd: CATTCAAAGGGAGGGCACAG
*DUE 3* Rev: AGCCGCTTGATGGACTTGTA
*P1* Fwd: CTTCCTGCCCGTAGTTCC
*P1* Rev: GCGCCCCTTTTCTTTCGG
*P2* Fwd: GAAAGAAAAGGGGCGCCG
*P2* Rev: GCTCTCGCGCACAAAGTT
*P3* Fwd: CAACTTTGTGCGCGAGAG
*P3* Rev: ACCCTCGCTCTCCAACAAAG
*P4* Fwd: CTTTGTTGGAGAGCGAGGGT
*P4* Rev: GAGGCTGGGCGATTCGTG

**Western blot**. Pelleted myoblasts were lysed in RIPA buffer (Sigma) supplemented with 1x cOmplete Protease Inhibitor Cocktail (Roche), followed by 40 min incubation with 125 U Benzonase (Sigma) at 4 ℃ with rotation. Protein quantification was performed using DC Protein Assay (BioRad). Protein was incubated with Laemmli Sample Buffer (BioRad) for 10 min at 95 ℃ then run at 80 V for 2 h on polyacrylamide pre-cast resolving gels (BioRad). Resolved proteins were transferred onto a Polyvinylidene Fluoride (PVDF) membrane using Transfer Buffer (1.5% Tris Base, 7.2% glycine in $H_2O$) for 2 h at 80 V at 4 ℃.

Membranes were blocked in 5% milk in TBS for 1 h at RT then incubated with primary antibodies (utrophin 1:200 (Santa Cruz, sc-33700), EZH2 1:500 (BD, 612666), Total H3 1:2,000 (Abcam, ab1791), GAPDH 1:2,000 (Abcam, ab8245), H3K27me3 1:750 (Abcam, mab6002)) in 5% milk in TBS overnight at 4 ℃. After three 5 min washes with TBS plus 1% Tween, blots were incubated with secondary antibodies (Goat anti-Rabbit Alexa Fluor 680 (Thermo Fisher, A21109) or Goat anti-Mouse Alexa Fluor 680 (Invitrogen, A-21057) at 1:5,000 in 5% milk in TBS for 2 h at RT with agitation. Membranes were washed three times for 10 min followed by detection using the LI-COR Odyssey imaging machine. Results were analysed using the Image Studio Lite software.

**Statistics and reproducibility**. Microsoft Excel was used for calculations and GraphPad Prism (version 8) used for statistical analysis and to generate graphs. Graphs show the mean of experimental replicates and standard error (SEM), with details provided in the figure legends. Multi-group comparisons were tested using one-way ANOVAs with Dunnett's or Sidak's correction for multiple comparisons. Pair-wise comparisons were tested using a paired *t*-test.

**Reporting summary**. Further information on research design is available in the Nature Portfolio Reporting Summary linked to this article.

## Data availability

All data generated during this study are included in the supplementary source data files (Supplementary data 1: numerical data for all graphs), and Supplementary Fig. 6 (full blots corresponding to Fig. 4f), or can be provided upon request.

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

## Acknowledgements

This work was funded by the Medical Research Council (MRC) (A.G.F. by MC_U120027516 and MC_UP_1605/12, M.M. by MC_UP_1605/11). B.F.R. was supported by funding from the British Heart Foundation (BHF) (REA grant RE/13/4/30184) and an ERDA award from the Institute of Clinical Sciences, Imperial College London. We would like to thank Professor Kay Davies (University of Oxford) for early discussions on the design of $Utrn^R$ reporter mice, Dr Anthony Uren (University of Plymouth) for his advice and support in the design of $Utrn^R$ and $Dmd^G$ reporter constructs and Dr Joaquim Pombo (MRC LMS) for his invaluable help with cryostat sectioning. For the purpose of open access, the author has applied a CC BY public copyright licence to any Author Accepted Manuscript version arising from this submission.

## Author contributions

H.J.G., B.F.R. and A.G.F. designed this study. H.J.G. and B.F.R. performed the majority of experiments and analysis. M.V.P. and A.G.F. designed the $Dmd^G$ and $Utrn^R$ reporter mouse lines. A.S. and M.V.P. provided support with animal work and IVIS imaging. J.M. performed OPT analysis. A.D. provided guidance on bioluminescent drug screening and writing. R.K.P. provided GSK drugs for screening. H.B. designed and generated $Dmd$ mutant myoblasts. M.M. provided support with animal work and experimental design. P.M.W.F. provided support with imaging and experimental design. The manuscript was written by A.G.F. and H.J.G. with editorial assistance from all authors. H.J.G. and B.F.R. made equivalent contributions to the study.

## Competing interests

The authors declare the following competing interests: R.K.P is an employee and shareholder of GSK. All other authors declare no competing interests.
