## [Peer review File · Communications Biology]

Reviewers' comments:

Reviewer #1 (Remarks to the Author):

This paper reports the generation of mouse reporters to visualise endogenous Dmd and Utrn gene expression simultaneously. The authors report in vivo data as well as the screening of myoblast lines with epigenetic modifiers. Their data support the view that EXH2 and ERK1/2 inhibitors increase utrophin levels and provide the resources for further screens to identify modulators of utrophin expression for potential clinical application for the therapy of DMD.

The paper describes the mouse Utrn expression which is consistent with the pattern reported previously using mRNA detection during embryonic development suggesting that the tools generated reflect the true expression for Utrn. The expression of both dystrophin and utrophin throughout development are reported which shows that utrophin expression is seen earlier than dystrophin. These data will be helpful in understanding the relationship between the expression of these two genes in fetal development and in the adult muscle.

Using data collected from myoblasts harvested from these mice, the authors show that increased utrophin is seen in myoblasts but not mature myotubes after treatment with drugs that are inhibitors of EXH2 or ERK1/2. This is important for the future clinical application of any inhibitors of EXH2 or ERK1/2.

The data presented support the hypothesis that combined inhibition of EZH2 and ERK1/2 might promote the precocious expression of differentiation associated properties by myoblasts which includes Utrn upregulation. The administration of drugs which target these pathways early on in the disease may well prove a fruitful strategy for the therapy of DMD. However it is puzzling that some drugs work but others do not when targeting the same pathway. e.g. GSK343 vs GSK503 which are extremely similar in profile, but one gives an increase in utrophin mRNA but the other does not (Figure 4c). Similarly the difference between GSK602 and GSK959. How do the authors explain this?

In summary, there is a lot of interest in increasing the levels of utrophin for the therapy of DMD. The data presented there provide novel insights into pathways which may be targeted in the future and provide the tools to screen for new drugs and will be of interest.

Two minor comments:

An screening cell lines has been previously reported which uses a knock in of a reporter into the utrophin locus (<https://www.sciencedirect.com/science/article/pii/S0223523421002804?via=ihub>) This should be referenced.

In the introduction they mention functional studies in the mouse mdx model which have shown that utrophin expression can alleviate the mdx phenotype and restore muscle function in a preclinical setting. They mean "prevent" as this has not been demonstrated in later stages of the disease in the mdx.

Mention should also be made of the dog utrophin data (Song et al Nat Med. 2019 Oct; 25(10): 1505–1511.)

Reviewer #2 (Remarks to the Author):

This study by Fisher and colleagues reports on the development of a new mouse model in which endogenous Dmd and Utrn gene expression can be simultaneously visualized in vivo, but most puzzlingly, they only use it to screen for compounds that increase Utrn expression in vitro. They report that PRC2 and ERK1/2 inhibitors increase Utrn expression, but the effects are only measured in vitro and the ms reports no data on the effect of these inhibitors on the dystrophy phenotypes of the mdx mouse model of DMD. Previously, several studies have reported on compounds with similar efficacies to increase Utrn expression in vitro, only to fail in clinical trial, or in preclinical studies. I might be more enthusiastic if they included data showing that the PRC2 and ERK1/2 inhibitors had significant

effects on ameliorating phenotypes in mdx mice like serum CK, CNF, muscle specific force and contraction-induced injury.

Reviewer #3 (Remarks to the Author):

The paper by Gleneadie reported the development of mice where the expression of dystrophin and utrophin can be simultaneously visualized by luciferase expression. The animal model is therefore of utmost importance for the researchers who work on different aspects of DMD and utrophin expression. With this model, the authors screened a number of molecules and identified a few that can upregulate utrophin expression in myoblasts. However, the upregulation of utrophin is limited to the myoblast only.

The experiments are sufficient to draw the conclusions made in the paper.

However, the detailed strategy for the construction of Luciferase knock-in is missing for dystrophin and utrophin; rather fig 1a and b are a bit confusing. If the boxes represent exons, then the position of Luc seems outside of the last exon. What do the lines between the boxes of T2A, CBG99Luc, or between T2A, RLuc, T2A and LacZ denote? It would be easy to understand, if the 3'UTR sequences of the dystrophin and utrophin with Luc, LacZ and T2A are shown, or the way the authors presented the strategy for Dlk1 in their previous paper (Fig 1a of Van de Pette et al.) is used here.

Point-by-Point Response to Reviewers

Reviewer #1 (Remarks to the Author):

This paper reports the generation of mouse reporters to visualise endogenous Dmd and Utrn gene expression simultaneously. The authors report in vivo data as well as the screening of myoblast lines with epigenetic modifiers. Their data support the view that EXH2 and ERK1/2 inhibitors increase utrophin levels and provide the resources for further screens to identify modulators of utrophin expression for potential clinical application for the therapy of DMD. The paper describes the mouse Utrn expression which is consistent with the pattern reported previously using mRNA detection during embryonic development suggesting that the tools generated reflect the true expression for Utrn. The expression of both dystrophin and utrophin throughout development are reported which shows that utrophin expression is seen earlier than dystrophin. These data will be helpful in understanding the relationship between the expression of these two genes in fetal development and in the adult muscle. Using data collected from myoblasts harvested from these mice, the authors show that increased utrophin is seen in myoblasts but not mature myotubes after treatment with drugs that are inhibitors of EXH2 or ERK1/2. This is important for the future clinical application of any inhibitors of EXH2 or ERK1/2.

The data presented support the hypothesis that combined inhibition of EZH2 and ERK1/2 might promote the precocious expression of differentiation associated properties by myoblasts which includes Utrn upregulation. The administration of drugs which target these pathways early on in the disease may well prove a fruitful strategy for the therapy of DMD. However it is puzzling that some drugs work but others do not when targeting the same pathway. e.g. GSK343 vs GSK503 which are extremely similar in profile, but one gives an increase in utrophin mRNA but the other does not (Figure 4c). Similarly the difference between GSK602 and GSK959. How do the authors explain this?

We thank the reviewer for their feedback and comments regarding the potential of targeting EZH2 and ERK1/2 pathways early in disease.

Regarding drugs that target the same pathway behaving differently in our assays.

Figure 4b shows that *Utrn*-derived bioluminescent signal is significantly increased following 24 hour treatment with GSK343 and GSK503 (EZH2 inhibitors), with GSK602 and GSK959 (bromodomain inhibitors) and with ERK pathway inhibitors (LY32, Ravox and LY3009). We have reworked this figure to highlight these results in yellow for ease of identification (relative to Figure 4c). Increased abundance of *Utrn* mRNA was detected in each of these samples (Figure 4c), with the exception of GSK343 (not statistically significant). Genetic experiments (conditional deletion) was used to confirm a role for EZH2 in regulating *Utrn* expression in myoblasts (Figures 4d-f).

Taken together, these results do not necessarily suggest any major differences in the behaviours of pairs of drugs targeting the same pathways – but do underscore that bioluminescence-derived detection of *Utrn* expression (flux) is a more sensitive tool for assessing *Utrn* expression than monitoring steady-state *Utrn* mRNA levels; RNA data is normalised to control genes and the resulting fold-changes in gene expression are less than those directly observed using bioluminescence.

In summary, there is a lot of interest in increasing the levels of utrophin for the therapy of DMD. The data presented there provide novel insights into pathways which may be targeted in the future and provide the tools to screen for new drugs and will be of interest.

We thank reviewer 1 for their encouraging words and look forward to sharing these new tools with the DMD community.

Two minor comments:

An screening cell lines has been previously reported which uses a knock in of a reporter into the utrophin locus

(<https://www.sciencedirect.com/science/article/pii/S0223523421002804?via=ihub>)

This should be referenced.

Many thanks for pointing this out, the reference has been incorporated in the revised discussion. The altered text is shown below, and highlighted in the resubmitted manuscript.

'In addition, bioluminescence-based cell reporters for *Utrn* are being successfully applied to screen for small molecule modulators of utrophin⁶⁶ (Line 54-58).

In the introduction they mention functional studies in the mouse mdx model which have shown that utrophin expression can alleviate the mdx phenotype and restore muscle function in a preclinical setting. They mean "prevent" as this has not been demonstrated in later stages of the disease in the mdx.

Mention should also be made of the dog utrophin data (Song et al Nat Med. 2019 Oct; 25(10): 1505–1511.)

Yes, this is a good point. We have amended the text accordingly and as requested, have added the dog utrophin data to the revised introduction. The altered text is shown below and highlighted in the resubmitted manuscript.

'Functional studies in the mouse *mdx* model of DMD¹⁹ have shown that utrophin expression can prevent development of the *mdx* phenotype and improve muscle function²⁰⁻²². Likewise, in a dystrophin-deficient golden retriever model, expression of a miniaturized utrophin transgene in neonatal or 7.5-week-old pups prevented development of myonecrosis²³ (Line 336-338).

Reviewer #2 (Remarks to the Author):

This study by Fisher and colleagues reports on the development of a new mouse model in which endogenous *Dmd* and *Utrn* gene expression can be simultaneously visualized *in vivo*, but most puzzlingly, they only use it to screen for compounds that increase *Utrn* expression *in vitro*. They report that PRC2 and ERK1/2 inhibitors increase *Utrn* expression, but the effects are only measured *in vitro* and the ms reports no data on the effect of these inhibitors on the dystrophy phenotypes of the *mdx* mouse model of DMD. Previously, several studies have reported on compounds with similar efficacies to increase *Utrn* expression *in vitro*, only to fail in clinical trial, or in preclinical studies. I might be more enthusiastic if they included data showing that the PRC2 and ERK1/2 inhibitors had significant effects on ameliorating phenotypes in *mdx* mice like serum CK, CNF, muscle specific force and contraction-induced injury.

We thank reviewer 2 for this feedback and share their frustration that many compounds identified *in vitro* have not made it to the clinic. Part of the problem may rest with the screening platforms and in part, with the animal models; as the *mdx* mouse model does not really recapitulate the phenotypes of Duchenne Muscular Dystrophy patients. In our view, addressing the efficacy of these drugs *in vivo* requires a further study, in addition, it will be important to see whether these drugs have similar impacts on human myoblasts.

To reiterate, our study provides new mechanistic insights on *Utrn* regulation and a novel screening platform and tools to aid in DMD drug discovery. We also demonstrate that PRC2 and ERK1/2 inhibitors alone or combined, increase *Utrn* expression in *Dmd*-mutant myoblasts; a prerequisite for further preclinical studies as it demonstrates that *Utrn* upregulation can be achieved in the absence of dystrophin.

Reviewer #3 (Remarks to the Author):

The paper by Gleneadie reported the development of mice where the expression of dystrophin and utrophin can be simultaneously visualized by luciferase expression. The animal model is therefore of utmost importance for the researchers who work on different aspects of DMD and utrophin expression. With this model, the authors screened a number of molecules and identified a few that can upregulate utrophin expression in myoblasts. However, the upregulation of utrophin is limited to the myoblast only.

The experiments are sufficient to draw the conclusions made in the paper.

We are grateful to reviewer 3 for this feedback.

However, the detailed strategy for the construction of Luciferase knock-in is missing for dystrophin and utrophin; rather fig 1a and b are a bit confusing. If the boxes represent exons, then the position of Luc seems outside of the last exon. What do the lines between the boxes of T2A, CBG99Luc, or between T2A, RLuc, T2A and LacZ denote? It would be easy to understand, if the 3'UTR sequences of the dystrophin and utrophin with Luc, LacZ and T2A are shown, or the way the authors presented the strategy for Dlk1 in their previous paper (Fig 1a of Van de Pette et al.) is used here.

We apologise for any lack of clarity here and thank the reviewer for bringing this to our attention. We have now re-worked and revised this figure to better represent the targeting strategy (revised Figures 1a and 1b). We hope that this has made the information clearer and more accessible for the reviewer as well as other readers.